# Automatic Radiobiological Comparison of Radiation Therapy Plans: An Application to Gastric Cancer

**DOI:** 10.3390/cancers14246098

**Published:** 2022-12-11

**Authors:** Michalis Mazonakis, Eleftherios Tzanis, Efrossyni Lyraraki, John Damilakis

**Affiliations:** 1Department of Medical Physics, Faculty of Medicine, University of Crete, 71003 Iraklion, Greece; 2Department of Radiation Oncology, University Hospital of Iraklion, 71110 Iraklion, Greece

**Keywords:** gastric cancer, 3D-CRT, IMRT, VMAT, treatment plan comparison, TCP, NTCP

## Abstract

**Simple Summary:**

The radiotherapy plans are usually judged using physical quantities, including dose and dose-volume values. Biological indices may also be applied for plan evaluation. A new software was developed for the quick and automatic calculation of tumor control probability (TCP) and normal tissue control probability (NTCP) from any radiation therapy plan and any fractionation scheme. Limited information has been published about the radiobiological metrics associated with gastric cancer irradiation. The new software was applied to radiobiologically compare photon plans for gastric malignancies generated with three-dimensional conformal radiotherapy (3D-CRT), intensity-modulated radiotherapy (IMRT) and volumetric modulated arc therapy (VMAT). The IMRT and VMAT plans provided higher TCPs than 3D-CRT. They also reduced the NTCPs and the risk of late adverse effects in most of the surrounding healthy organs compared to 3D-CRT. The presented results are useful for the plan optimization and choice of the appropriate radiotherapy technique for gastric cancer.

**Abstract:**

(1) Aim: This study was conducted to radiobiologically compare radiotherapy plans for gastric cancer with a newly developed software tool. (2) Methods: Treatment planning was performed on two computational phantoms simulating adult male and female patients. Three-dimensional conformal radiotherapy (3D-CRT), intensity-modulated radiation therapy (IMRT) and volumetric modulated arc therapy (VMAT) plans for gastric cancer were generated with three-photon beam energies. The equivalent uniform dose (EUD), tumor control probability (TCP) of the target and normal tissue control probability (NTCP) of eight different critical organs were calculated. A new software was employed for these calculations using the EUD-based model and dose-volume-histogram data. (3) Results: The IMRT and VMAT plan led to TCPs of 51.3–51.5%, whereas 3D-CRT gave values up to 50.2%. The intensity-modulated techniques resulted in NTCPs of (5.3 × 10^−6^–3.3 × 10^−1^)%. The corresponding NTCPs from 3D-CRT were (3.4 × 10^−7^–7.4 × 10^−1^)%. The above biological indices were automatically calculated in less than 40 s with the software. (4) Conclusions: The direct and quick radiobiological evaluation of radiotherapy plans is feasible using the new software tool. The IMRT and VMAT reduced the probability of the appearance of late effects in most of the surrounding critical organs and slightly increased the TCP compared to 3D-CRT.

## 1. Introduction

Gastric cancer is the fifth most common malignant disease worldwide, with more than one million new cases per year, according to recent statistics from 185 countries [1]. Gastric carcinoma is also the fourth leading cause of cancer mortality globally, following lung, colorectal and liver cancers [1]. Complete tumor and regional lymph node resection is the primary treatment approach for patients with stomach cancer [2]. The curative or palliative surgical resection may be applied in 50–60% of these subjects during the first disease stage [2]. Macdonald et al. [3] showed that postoperative chemoradiotherapy for gastric cancer significantly improves overall and relapse-free survival compared to those associated only with surgery.

Adjuvant radiation therapy administered with concurrent chemotherapy is currently recommended for patients suffering from gastric cancer [4]. Radiation-related toxicity remains a concern for these patients subjected to the above treatment option [3,4]. The target volume in gastric cancer usually has large dimensions [5], and it is surrounded by many healthy organs with high radiosensitivities, such as kidneys, liver, heart, bowel and lungs. Special consideration needs to be given in the selection of the proper radiation therapy plan, ensuring adequate tumor control with minimal adverse events. The radiobiological optimization of treatment plans using biologically related models should supersede the well-known plan evaluation based on physical dose quantities [6]. Limited data have been published about the radiobiological metrics associated with the therapeutic irradiation of gastric cancer. Mondlane et al. [7] found that proton therapy reduced the normal tissue control probability (NTCP) of the left kidney with respect to volumetric modulated arc therapy (VMAT) with 6 MV photons. Photon and proton irradiation led to similar NTCPs for all other organs-at-risk (OARs). Sharfo et al. [8] showed that the automated VMAT plans result in lower NTCPs of kidneys and liver compared to manual VMAT planning with 10 MV photons. To our knowledge, no attempts have been made to investigate the impact of radiation therapy technique and photon beam energy on the tumor control probability (TCP) and NTCP related to gastric cancer irradiation.

The objectives of this study were (a) to determine the TCP and NTCP from three-dimensional conformal radiotherapy (3D-CRT), intensity modulated radiation therapy (IMRT) and VMAT for gastric cancer with different photon beam energies, and (b) to develop a new software package providing automatic calculation of the above radiobiological parameters derived from the treatment plans.

## 2. Materials and Methods

### 2.1. XCAT Male and Female Phantoms

This work was carried out using 4D extended cardiac-torso (XCAT) computational phantoms. Analytical whole-body anatomies of male and female adults were defined by Segars et al. [9] with non-uniform rational B-splines. The bones, muscles and vessels of the human body, together with all internal organs, were modeled in the XCAT phantoms [9]. The height and weight of the XCAT phantom simulating a male patient were 1761 mm and 81 kg, respectively. The corresponding dimensions for the female phantom were 1627 mm and 66 kg. The XCAT phantoms have already been used for treatment planning and dosimetric purposes in radiation therapy [10,11,12].

### 2.2. Treatment Planning for Gastric Cancer

The dicom XCAT images were compatible with Monaco treatment planning system (Elekta Instrument AB, Sweden). A series of 3-mm CT images depicting the anatomy of each XCAT phantom was used for contouring. The phantoms were in supine position. A radiation oncologist manually contoured the clinical target volume (CTV) and the planning target volume (PTV) for gastric cancer treatment. Manual delineation was performed for the following organs-at-risk (OARs): left and right lungs, heart, left and right kidneys, liver, bowel and spinal cord. The treatment plans were created by an experienced medical physicist for an Infinity linear accelerator (Elekta Instrument AB, Sweden) producing 6, 10 and 15 MV photons. The prescription dose was 45 Gy, given in 25 fractions.

Three-dimensional conformal radiotherapy (3D-CRT) plans were generated using the field-in-field technique. Three different plans consisting of four treatment fields at the gantry angles of 0°, 90°, 180° and 270° were generated using 6, 10 and 15 MV photon beams. No more than four segments were used for each field. The IMRT plans included a seven-field arrangement at the following angles: 0°, 51°, 102°, 153°, 204°, 255° and 306° [13]. The VMAT plans were designed using two complete arcs in opposite directions. The above VMAT planning strategy is used for abdominopelvic tumors in our department [14,15]. Six and ten MV X-rays were used for IMRT and VMAT plans. For 3D-CRT plans, 95% of the PTV had to be covered by at least the 95% of the prescribed dose. The corresponding goal for plans with intensity-modulated beams was V_42_._75Gy_ ≥ 99%. The previously reported dose constraints for lungs, heart, kidneys, liver, bowel and spinal cord were used for generation of the above treatment plans [4]. The constraints are shown in Table 1.

The average dose (D_av_) received by each OAR and the parameters V_nGy_ denoting the organ volume exceeding n Gy were recorded from the dose-volume histograms (DVHs). The maximum dose (D_max_) to spinal cord was also taken from the DVHs. For the PTV, the homogeneity index (HI) and the conformation number (CN) were determined. The HI was found with the following equation:(1)$$\mathrm{HI}=\frac{D_{5\%}}{D_{95\%}}$$
where D_5%_ and D_95%_ are the radiation doses to 5% and 95% of the PTV. The CN was calculated as follows:(2)$$\mathrm{CN}=\frac{{{(\mathrm{PTV}}_{\mathrm{PD}})}^{2}}{{\mathrm{PTV}\;\times \;V}_{\mathrm{PD}}}$$
where PTV_PD_ is the target volume covered by the prescribed dose, PTV is the volume of the target and V_PD_ is the whole volume of the phantom receiving the prescribed dose.

### 2.3. TCP and NTCP Calculations

The equivalent uniform dose (EUD) model was used for calculating the TCP NTCPs [16,17]. These calculations were performed for the seven different radiation therapy plans for gastric cancer generated on a phantom representing an average male or female patient. The EUD was found with the formula:(3)$$\mathrm{EUD}={\left[\underset{i=1}{\sum }\left(V_{i}D_{i}^{\alpha }\right)\right]}^{\frac{1}{\alpha }}$$
where V_i_ is the partial volume i absorbing a radiation dose equal to D_i_ and *α* is a unitless factor associated with OAR or target. The V_i_ and D_i_ inserted in Equation (3) were obtained by the DVH of each structure of interest. For fraction doses other than 2 Gy, the biological equivalent physical dose (EQD_i_) needs to be inserted in Equation (3). The EQD_i_ was calculated as follows:(4)EQDi=Di(αβ+Df)(αβ+2)
where αβ is the tissue-dependent linear quadratic parameter for the irradiated tissue and $D_{f}$ is the tumor dose per fraction.

The TCP was calculated as follows:(5)$$\mathrm{TCP}={\left[1+{\left(\frac{{\mathrm{TCD}}_{50}}{\mathrm{EUD}}\right)}^{{4\gamma }_{50}}\right]}^{-1}$$
where TCD_50_ is the radiation dose to tumor, resulting in control of 50% of the cancer cells, and γ_50_ is a model parameter related to each tumor type. The NTCP for lungs, heart, kidneys, liver, bowel and spinal cord was found with the following equation:(6)$$\mathrm{NTCP}={\left[1+{\left(\frac{{\mathrm{TD}}_{50}}{\mathrm{EUD}}\right)}^{{4\gamma }_{50}}\right]}^{-1}$$
where TD_50_ is the dose tolerance leading to a 50% complication rate.

### 2.4. Development of the Software Tool

A new software package was developed to facilitate the radiobiological evaluation of treatment plans. Scripting was performed with python 3.8 programming language. The algorithm of the software incorporates Equations (3)–(6). The required parameters TD_50_, α, α/β and γ_50_ for the NTCP calculation of twenty-three different critical organs were taken from the literature [17,18,19,20,21,22,23,24], and they were inserted into the software. The above parameters are summarized in Table 2. The respective parameters for twelve macroscopic tumors and eleven microscopic malignancies derived from Okunieff et al. [25] were also incorporated into the new tool for plan analysis. To minimize the preprocessing time, the algorithm was developed to retrieve V_i_ and D_i_ values automatically from differential DVH text files exported from the Monaco treatment planning system. The graphical user interface (GUI) was developed with the tkinter module [26].

Figure 1, Figure 2 and Figure 3 indicatively show the software’s architecture and workflow. On the home screen, the user has the option to activate the TCP or NTCP calculator (Figure 1). By selecting the NTCP button, the NTCP’s GUI appears and the user can select the OAR under investigation, define the tumor dose per fraction in Gy or cGy and insert the relevant DVH file (Figure 2). By selecting the ‘Estimate EUD and NTCP’ button, the algorithm automatically retrieves the required calculation parameters, performs all necessary computations and then presents the EUD and NTCP values together with the relevant endpoint in a new window. Moreover, the software gives the option to export and save a text file which may include information about the patient name, radiotherapy technique, total tumor dose, organ-specific parameters employed for software calculation, as well as the resultant EUD and NTCP values (Figure 2).

The TCP’s GUI may be presented by selecting the TCP button in the initial window of Figure 3. The user has to select whether this calculation is about a microscopic or macroscopic malignant disease. Then, the user defines the tumor site under examination, tumor dose per fraction in Gy or cGy, disease stage and imports the DVH text file. Furthermore, the software allows the user to calculate the TCP for tumors not considered in our calculator. In this case, the user has to import the appropriate values for TCD_50_, α, α/β and γ_50_. The results of the TCP calculator can be easily exported in a similar way to that described for NTCP calculations.

The accuracy of the newly developed software tool was investigated in the current study. The TCP and NTCPs of eight different critical organs associated with a 6 MV IMRT plan for gastric cancer generated on a phantom simulating a female patient were calculated with the aid of the software. The obtained results were directly compared with the manual calculations of the aforementioned radiobiological parameters.

## 3. Results

### 3.1. Dosimetric Comparison of Treatment Plans

The isodose curves associated with 3D-CRT, IMRT and VMAT plans for gastric cancer are presented in Figure 4. The dose parameters for the PTV and eight different OARs derived from treatment plans on XCAT phantoms representing an adult male and female are shown in Table 3 and Table 4, respectively. The V_42_._75Gy_ for the PTV in 3D-CRT plans varied from 95.8% to 98.3% by the beam energy and the phantom used. The respective value for plans with intensity-modulated beams was at least 99.9%. IMRT plans led to HI and CN values of 1.03–1.04 and 0.84–0.87, respectively. The corresponding parameters from VMAT were similar and equal to 1.03–1.04 and 0.85–0.87. The 3D-CRT plans led to a HI of 1.08–1.10, whereas the CN was 0.35–0.61.

The D_av_ received by the left lung, right lung, heart, left kidney, right kidney and liver from 3D-CRT plans for gastric cancer generated on both humanoid phantoms was 4.1–6.8 Gy, 2.3–4.6 Gy, 9.4–17.6 Gy, 12.0–15.7 Gy, 9.9–12.6 Gy and 21.5–22.6 Gy, respectively. The corresponding doses from IMRT and VMAT were 5.5–8.4 Gy, 4.0–7.0 Gy, 10.2–16.5 Gy, 10.6–12.8 Gy, 9.3–11.0 Gy and 19.5–21.2 Gy, respectively. The D_max_ values to the spinal cord from 3D-CRT, IMRT and VMAT were 31.1–35.6 Gy, 33.1–37.7 Gy and 32.3–36.7 Gy, respectively. The V_10Gy_, V_20Gy_ and V_30Gy_ for both lungs from 3D-CRT were found to be smaller than those from IMRT and VMAT plans. The opposite result was found for the V_i_ parameters of right and left kidneys, liver and bowel. The V_30Gy_ for the heart associated with conformal radiotherapy of a female patient with gastric cancer was lower than that from IMRT and VMAT. The above was not observed for treatment plans generated on the XCAT phantom representing an adult male.

### 3.2. Radiobiological Comparison of Treatment Plans

No difference was detected between the manual calculations of both TCP and NTCP and the respective software-based calculations. The time for any software calculation took less than 40 s. The TCP and NTCP of an adult male undergoing radiation therapy for gastric cancer, as obtained by the new software tool, are presented in Table 5. The radiobiological parameters for an irradiated female with stomach carcinoma are shown in Table 6. The TCP range from 3D-CRT plans created on both phantoms was 48.5–50.2%, depending upon the photon beam energy used. The TCPs from IMRT and VMAT varied from 51.3 to 51.5%. The NTCP for lungs, heart, kidneys, liver, bowel and spinal cord due to 3D-CRT of male patients with gastric cancer was (4.8 × 10^−5^–7.4 × 10^−1^)%. The corresponding ranges from IMRT and VMAT were (9.2 × 10^−6^–3.3 × 10^−1^)% and (1.3 × 10^−5^–2.7 × 10^−1^)%. The NTCPs derived from 3D-CRT, IMRT and VMAT plans created on a phantom simulating an average female were (3.4 × 10^−7^–5.2 × 10^−1^)%, (5.3 × 10^−6^–2.9 × 10^−1^)% and (5.9 × 10^−6^–2.4 × 10^−1^)%, respectively.

## 4. Discussion

The quality of any particular radiotherapy plan and the subsequent comparison of plans usually relies on radiation dose and dose-volume parameters. The report of the therapy physics committee of the American Association of Physicists in Medicine previously suggested the use of biologically-based models for treatment planning [6]. The TCP and NTCP indices may reflect more closely the clinical outcome of radiation therapy compared to physical dose-volume quantities [6]. The above radiobiological indices have been widely employed for the analysis and evaluation of radiation therapy plans [27,28,29]. However, many treatment planning systems can not directly provide TCP and NTCP calculations. The determination of these quantities on the basis of the EUD model requires the extraction of DVHs from the planning systems and, then, the combination of histogram data with mathematical equations and model parameters. This procedure takes considerable time making its use to be rather difficult in everyday clinical practice.

This study introduced a new software tool for calculating the TCP and NTCP from radiation therapy plans. This tool was not limited to the needs of the present study related to gastric cancer and the surrounding OARs. It was designed to give calculations of radiobiological parameters for different tumor sites and different critical organs. Moreover, its design enabled the TCP and NTCP determination for any fractionation schedule. The DVH data of the target volume or any OAR, as obtained by the treatment planning system, was directly introduced into the software tool without any processing or modification. Organ- and tumor-specific parameters were introduced into the software environment to facilitate the EUD calculation and the subsequent determination of the TCP and/or NTCP. The user had to simply define the OAR or tumor site and the dose per fraction. The accuracy of the software results was verified against manual calculations of the above biological indices. The proposed software tool may be routinely used as a clinical aid in the radiobiological comparison of any treatment plans. The user intervention in this process is minimal. The tool may directly give quick and automatic calculations of TCP and NTCP values. Further research is needed to investigate the application of the software tool to centers equipped with treatment planning systems different from that used in this work.

The newly developed software tool was applied for the radiobiological comparison of photon plans for gastric cancer. The target volume in radiotherapy for gastric malignancies has multi-concave shapes, and its size may vary widely from patient to patient. Jansen et al. [30] found a PTV range from 634 cm^3^ to 1677 cm^3^. The analysis of treatment plans obtained by consecutive gastric cancer patients undergoing radiation therapy may not lead to representative dose results for the average subject. The previously reported guidelines of radiotherapy treatment planning [31] pointed out that some planning studies may benefit from the in-depth and profound examination of a few representative cases. The treatment plans of this study were created on two different computational XCAT phantoms simulating average adult male and female patients. The anatomies of the XCAT phantoms were properly adjusted to represent the 50th percentile of United States adult subjects aged 18–64 years [9]. Data from ICRP publication 89 [32] dealing with anatomical and physiological values of reference individuals were employed to determine the organ volumes of both phantoms.

Three dimensional-CRT plans were generated with 6, 10 and 15 MV X-rays on both XCAT phantoms. Photon beam energies of 6 MV and 10 MV were used for the IMRT and VMAT planning. The generated treatment plans were considered clinically acceptable, and they fulfilled the dose constraints for the examined OARs. The IMRT and VMAT plan led to almost the same TCP values, which were always higher than those related to 3D-CRT. The TCP difference between the intensity-modulated treatment techniques and conformal radiotherapy was found to be 4.0–6.0% for females treated for the gastric difference. The minimum difference for irradiated males was 2.2%. The calculated TCP values were consistent with those reported by Mehri-Kakavand et al. [19], who examined patients subjected to radiotherapy for gastro-esophageal junction cancer.

The NTCP values for the spinal cord and the critical abdominal organs, including left and right kidneys, liver and small bowel associated with IMRT and VMAT, were systematically lower than those from 3D-CRT for gastric cancer irrespective of the patient’s gender and the beam energy employed. Conventional conformal radiation therapy on a phantom simulating an adult female resulted in smaller NTCPs for both lungs and heart with respect to advanced intensity modulated treatment methods. Similar results were found only for the left and right lungs based on the analysis of treatment plans created on a phantom representing a typical male. It should be noted that the NTCP values derived from all treatment plans for gastric cancer created on the two different XCAT phantoms were found to be small. The vast majority of these values were lower than 0.01%. One exception was the NTCP of the liver, which varied from 0.24% to 0.74%. The NTCP of the left kidney for males subjected to 3D-CRT was also more than 0.03%.

Comparing the biological indices from plans generated with different photon beam energies, useful conclusions were drawn about the optimal energy selection for radiotherapy for gastric cancer. The TCP and NTCP from 3D-CRT with 15 MV photons were lower than those associated with treatment plans created with both 6 and 10 MV X-rays. The photon beam energy had no effect on the TCP values derived from both IMRT and VMAT plans. However, the use of 10 MV instead of 6 MV photons for IMRT and VMAT for gastric cancer reduced the NTCP to five to six out of eight OARs examined in this work.

## 5. Conclusions

In conclusion, user-friendly software was developed for the calculation of TCP and NTCP values. The time efficiency and accuracy of the software make it a useful tool for the implementation of the radiobiological evaluation of radiotherapy treatment plans in everyday clinical practice. A radiobiological comparison of 3D-CRT, IMRT and VMAT plans for gastric cancer was performed with the new software. Both IMRT and VMAT resulted in slightly higher TCP values compared to 3D-CRT. The intensity-modulated treatment techniques were also superior to conventional irradiation in terms of dose homogeneity and conformity. The NTCPs of all the surrounding abdominal organs and spinal cord associated with IMRT and VMAT were lower than those from 3D-CRT plans. The above was not observed for the left and right lungs. The increase of the photon beam energy reduced the NTCP for most of the OARs under investigation, irrespective of the treatment technique and the patient’s gender. The results from the radiobiological comparison of treatment plans provide useful information about the late effects of irradiation, and they can be used in the selection of the appropriate radiotherapy technique for gastric cancer patients.

## Figures and Tables

**Figure 1 cancers-14-06098-f001:**
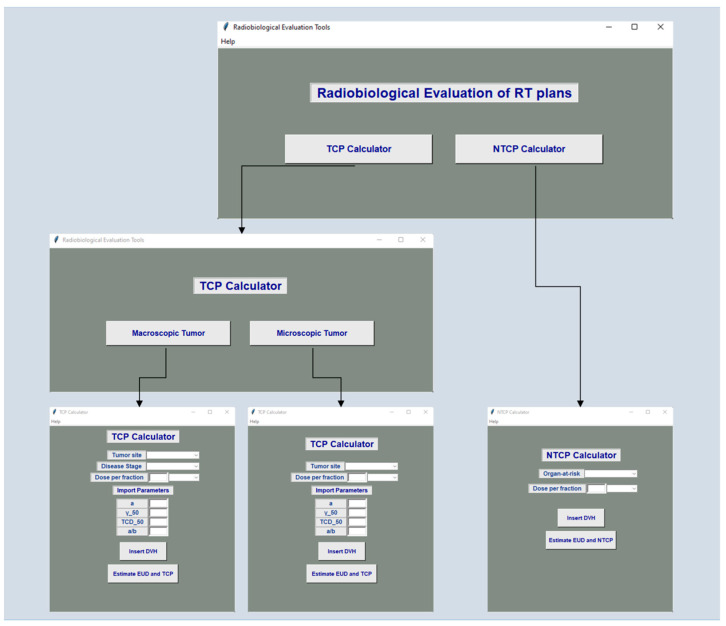
Architecture of the software tool for the radiobiological evaluation of radiation therapy plans.

**Figure 2 cancers-14-06098-f002:**
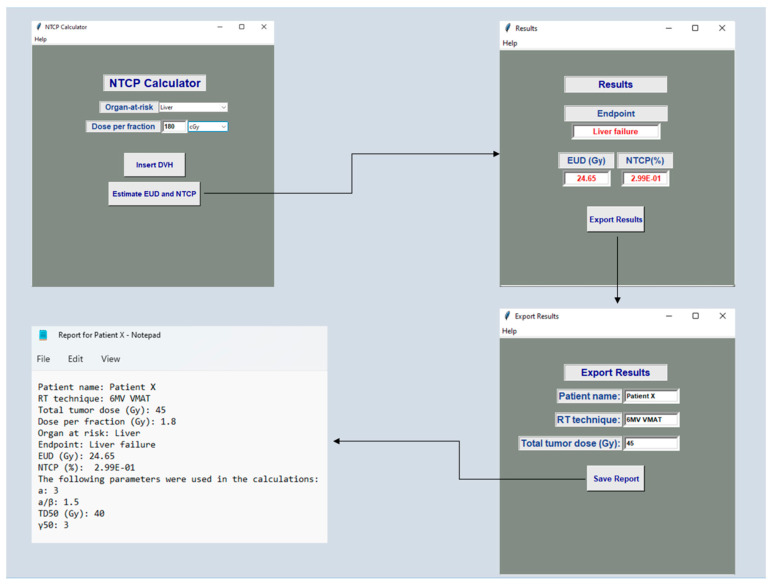
Software screen windows presenting the NTCP calculation and the extraction of the relevant report. This report is about the NTCP of the liver from a VMAT plan delivering 45 Gy with 6 MV photons.

**Figure 3 cancers-14-06098-f003:**
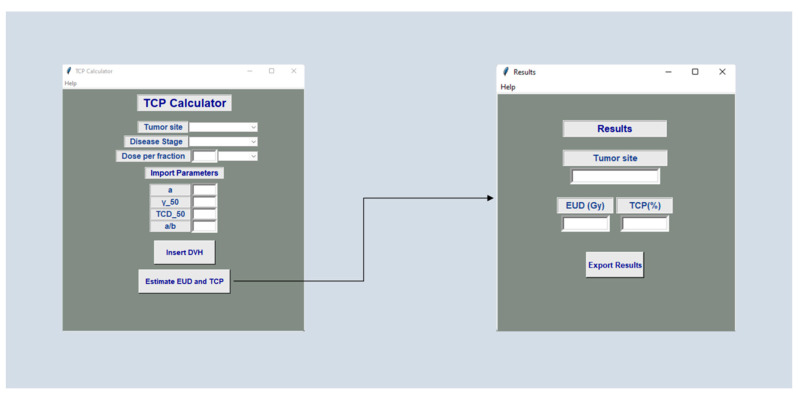
Software screen windows presenting the TCP calculation.

**Figure 4 cancers-14-06098-f004:**
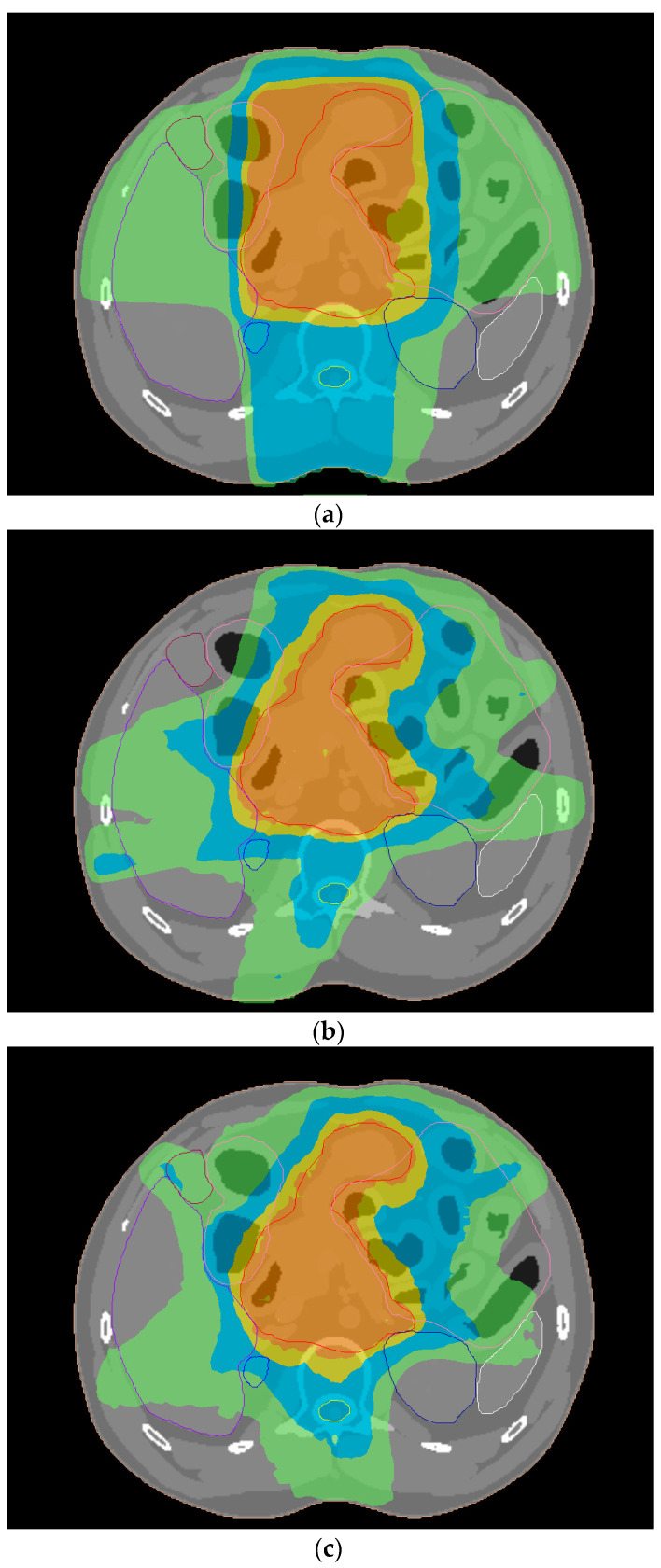
A color wash display of the PTV isodose distribution in an axial CT slice of a phantom simulating a male patient derived from (**a**) 3D-CRT, (**b**) IMRT and (**c**) VMAT plans with 6 MV photons. The isodoses of 100, 90, 65 and 45% are shown in orange, yellow, blue and green colors, respectively. The PTV is presented with the red contour.

**Table 1 cancers-14-06098-t001:** Dose constraints used for the generation of 3D-CRT, IMRT and VMAT plans for gastric cancer.

Organ-at-Risk	Parameter
Lungs	D_av_ < 20 Gy
	V_30Gy_ < 15%
	V_20Gy_ < 20%
	V_10Gy_ < 40%
Heart	D_av_ < 30 Gy
	V_30Gy_ < 30%
Kidneys	D_av_ < 18 Gy
	V_20Gy_ < 33%
Liver	D_av_ < 25 Gy
	V_30Gy_ < 33%
Bowel	V_45Gy_ < 195 cc
Spinal cord	D_max_ < 45 Gy

D_av_, average dose; V_nGy_, organ volume receiving more than n Gy; D_max_, maximum dose.

**Table 2 cancers-14-06098-t002:** Parameters introduced into the software for NTCP calculation.

Organ-at-Risk	α	γ_50_	TD_50_ (Gy)	α/β (Gy)	Endpoint
Brain stem	7	3	65	3	Necrosis
Parotid	0.5	4	46	2	Xerostomia
Ear (mid/ext)	31	3	40	10	Acute serious otitis
Ear (mid/ext)	31	4	65	3	Chronic serious otitis
TMJ	14	4	72	3	Limited joint function
Larynx	12.5	4	70	3.8	Laryngeal edema
Mandible	14	4	72	3	Limited joint function
Optic chiasm	25	3	65	3	Blindness
Optic nerve	25	3	65	3	Blindness
Eye lens	3	1	18	1.2	Cataract
Cochlea	31	3	65	3	Chronic serious otitis
Brain	5	3	60	2.1	Necrosis
Lung	1	2	24.5	3	Pneumonitis
Heart	3	3	50	2.5	Pericarditis
Liver	3	3	40	1.5	Liver failure
Kidney	1	3	28	2.5	Nephritis
Bowel	6	4	55	3	Obstruction
Stomach	14	3	65	5	Perforation
Esophagus	19	4	68	3	Perforation
Rectum	8	4	80	3.9	Necrosis/Stenosis/fistula
Bladder	2	4	80	8	Bladder contracture/volume loss
Femoral heads	4	4	65	0.85	Necrosis
Spinal cord	7.4	4	66.5	3	Myelitis/necrosis

**Table 3 cancers-14-06098-t003:** Dosimetric parameters for the PTV and OARs derived from radiation therapy plans for gastric cancer on a computational phantom representing an adult male patient.

Structure	Parameter	3D-CRT			IMRT		VMAT	
		6 MV	10 MV	15 MV	6 MV	10 MV	6 MV	10 MV
PTV	V_42_._75Gy_(%)	98.0	98.3	97.3	100.0	99.9	99.9	100.0
	HI	1.10	1.09	1.09	1.03	1.04	1.04	1.03
	CN	0.55	0.61	0.58	0.86	0.87	0.87	0.87
Left Lung	V_10Gy_(%)	21.5	21.5	21.3	29.3	31.3	28.3	27.5
	V2_0Gy_(%)	13.0	12.4	11.7	13.8	13.5	12.4	11.9
	V_30Gy_(%)	6.0	5.9	5.7	7.0	6.8	6.6	6.1
	D_av_ (Gy)	6.8	6.6	6.4	8.4	8.2	8.1	7.9
Right Lung	V_10Gy_(%)	13.6	13.5	13.3	24.5	22.8	27.4	27.9
	V_20Gy_(%)	6.9	5.9	4.8	7.6	7.6	7.7	8.1
	V_30Gy_(%)	1.1	1.1	1.1	2.9	2.8	2.9	3.0
	D_av_ (Gy)	4.6	4.4	4.2	6.8	6.5	7.0	6.8
Heart	V_30Gy_(%)	12.9	13.2	11.2	8.2	8.2	8.3	7.9
	D_av_ (Gy)	17.6	17.5	17.2	16.5	16.4	16.2	16.0
Left Kidney	V_20Gy_(%)	26.0	25.4	24.8	15.4	14.6	14.4	14.0
	D_av_ (Gy)	15.7	15.5	15.1	12.2	11.3	12.8	12.6
Right Kidney	V_20Gy_(%)	21.7	20.2	16.9	15.5	15.2	10.5	10.8
	D_av_ (Gy)	12.6	12.4	12.0	10.5	9.9	11.0	10.9
Liver	V_30Gy_(%)	20.4	20.4	20.3	19.2	20.0	17.7	18.2
	D_av_ (Gy)	22.6	22.4	21.9	20.8	21.2	21.1	21.0
Bowel	V_45Gy_ (cc)	151.8	165.7	134.4	69.7	68.2	68.2	71.7
Spinal cord	D_max_(Gy)	35.6	35.1	34.5	36.0	37.7	36.3	36.7

PTV, planning target volume; OAR, organ-at-risk; 3D-CRT, three-dimensional conformal radiotherapy; IMRT, intensity modulated radiation therapy; VMAT, volumetric modulated arc therapy; D_av_, average dose; V_nGy_, organ volume receiving more than n Gy; D_max_, maximum dose.

**Table 4 cancers-14-06098-t004:** Dosimetric parameters for the PTV and OARs derived from radiation therapy plans for gastric cancer on a computational phantom representing an adult female patient.

Structure	Parameter	3D-CRT			IMRT		VMAT	
		6 MV	10 MV	15 MV	6 MV	10 MV	6 MV	10 MV
PTV	V_42_._75Gy_(%)	96.7	97.4	95.8	100.0	100.0	99.9	99.9
	HI	1.09	1.08	1.08	1.03	1.03	1.04	1.04
	CN	0.35	0.52	0.44	0.84	0.84	0.85	0.86
Left Lung	V_10Gy_(%)	12.3	12.4	12.3	17.9	17.9	18.2	18.1
	V_20Gy_(%)	7.2	7.2	7.1	10.8	10.2	10.5	9.5
	V_30Gy_(%)	3.9	3.8	3.4	5.7	5.5	5.4	5.2
	D_av_ (Gy)	4.3	4.3	4.1	5.8	5.6	5.7	5.5
Right Lung	V_10Gy_(%)	5.8	5.8	5.7	13.0	12.6	14.3	14.1
	V_20Gy_(%)	1.7	1.6	1.5	5.1	4.8	5.7	6.0
	V_30Gy_(%)	0.4	0.4	0.4	2.0	1.7	1.9	1.9
	D_av_ (Gy)	2.5	2.4	2.3	4.2	4.0	4.4	4.4
Heart	V_30Gy_(%)	3.7	3.9	3.3	5.4	5.7	6.0	5.4
	D_av_ (Gy)	9.9	9.7	9.4	10.9	10.8	10.9	10.2
Left Kidney	V_20Gy_(%)	20.9	20.9	20.1	11.7	11.3	10.2	9.1
	D_av_ (Gy)	12.3	12.3	12.0	10.8	10.6	11.6	11.4
Right Kidney	V_20Gy_(%)	18.8	18.5	17.9	15.3	15.6	10.5	10.7
	D_av_ (Gy)	10.0	9.9	9.9	9.6	9.3	9.8	9.8
Liver	V_30Gy_(%)	18.7	19.0	18.6	18.4	18.3	17.6	17.6
	D_av_ (Gy)	22.1	22.0	21.5	19.8	19.5	19.9	19.9
Bowel	V_45Gy_ (cc)	85.4	94.7	69.6	52.7	52.2	52.3	50.8
Spinal cord	D_max_(Gy)	31.7	31.1	30.4	33.1	33.3	32.3	35.8

PTV, planning target volume; OAR, organ-at-risk; 3D-CRT, three-dimensional conformal radiotherapy; IMRT, intensity modulated radiation therapy; VMAT, volumetric modulated arc therapy; D_av_, average dose; V_nGy_, organ volume receiving more than n Gy; D_max_, maximum dose.

**Table 5 cancers-14-06098-t005:** TCP and NTCP calculations derived from radiation therapy plans for gastric cancer on a computational phantom representing an adult male patient.

Structure	Parameter	3D-CRT			IMRT		VMAT	
		6 MV	10 MV	15 MV	6 MV	10 MV	6 MV	10 MV
PTV	TCP (%)	50.0	50.2	49.7	51.4	51.3	51.3	51.3
Left Lung	NTCP (%)	2.4 × 10^−3^	2.0 × 10^−3^	1.5 × 10^−3^	1.3 × 10^−2^	1.2 × 10^−2^	9.5 × 10^−3^	7.5 × 10^−3^
Right Lung	NTCP (%)	9.9 × 10^−5^	7.2 × 10^−5^	4.8 × 10^−5^	2.4 × 10^−3^	1.6 × 10^−3^	3.0 × 10^−3^	2.6 × 10^−3^
Heart	NTCP (%)	9.2 × 10^−3^	5.6 × 10^−3^	4.8 × 10^−3^	1.6 × 10^−3^	1.5 × 10^−3^	1.4 × 10^−3^	1.2 × 10^−3^
Left Kidney	NTCP (%)	5.1 × 10^−2^	4.5 × 10^−2^	3.4 × 10^−2^	2.3 × 10^−3^	9.3 × 10^−4^	4.1 × 10^−3^	3.5 × 10^−3^
Right Kidney	NTCP (%)	3.6 × 10^−3^	2.9 × 10^−3^	2.2 × 10^−3^	3.5 × 10^−4^	1.7 × 10^−4^	6.7 × 10^−4^	6.7 × 10^−4^
Liver	NTCP (%)	7.4 × 10^−1^	7.3 × 10^−1^	6.2 × 10^−1^	3.0 × 10^−1^	3.3 × 10^−1^	2.7 × 10^−1^	2.7 × 10^−1^
Bowel	NTCP (%)	6.2 × 10^−3^	6.5 × 10^−3^	5.2 × 10^−3^	3.8 × 10^−3^	3.8 × 10^−3^	3.6 × 10^−3^	3.8 × 10^−3^
Spinal cord	NTCP (%)	9.1 × 10^−5^	7.2 × 10^−5^	5.0 × 10^−5^	9.2 × 10^−6^	1.4 × 10^−5^	1.5 × 10^−5^	1.3 × 10^−5^

TCP, tumor control probability; NTCP, normal tissue control probability; PTV, planning target volume; 3D-CRT, three-dimensional conformal radiotherapy; IMRT, intensity modulated radiation therapy; VMAT, volumetric modulated arc therapy.

**Table 6 cancers-14-06098-t006:** TCP and NTCP calculations derived from radiation therapy plans for gastric cancer on a computational phantom representing an adult female patient.

Structure	Parameter	3D-CRT			IMRT		VMAT	
		6 MV	10 MV	15 MV	6 MV	10 MV	6 MV	10 MV
PTV	TCP (%)	48.7	49.3	48.5	51.5	51.4	51.4	51.3
Left Lung	NTCP (%)	6.5 × 10^−5^	5.4 × 10^−5^	4.3 × 10^−5^	6.2 × 10^−4^	5.0 × 10^−4^	6.1 × 10^−4^	4.1 × 10^−4^
Right Lung	NTCP (%)	8.3 × 10^−7^	5.5 × 10^−7^	3.4 × 10^−7^	5.1 × 10^−5^	3.4 × 10^−5^	6.8 × 10^−5^	6.8 × 10^−5^
Heart	NTCP (%)	1.4 × 10^−4^	1.5 × 10^−4^	1.3 × 10^−4^	1.9 × 10^−4^	2.0 × 10^−4^	2.1 × 10^−4^	1.7 × 10^−4^
Left Kidney	NTCP (%)	3.1 × 10^−3^	2.7 × 10^−3^	1.8 × 10^−3^	5.3 × 10^−4^	4.1 × 10^−4^	1.2 × 10^−3^	1.1 × 10^−3^
Right Kidney	NTCP (%)	2.1 × 10^−4^	1.9 × 10^−4^	1.8 × 10^−4^	9.1 × 10^−5^	7.7 × 10^−5^	1.4 × 10^−4^	1.6 × 10^−4^
Liver	NTCP (%)	5.1 × 10^−1^	5.2 × 10^−1^	4.5 × 10^−1^	2.9 × 10^−1^	2.7 × 10^−1^	2.4 × 10^−1^	2.4 × 10^−1^
Bowel	NTCP (%)	6.2 × 10^−3^	6.4 × 10^−3^	5.2 × 10^−3^	4.3 × 10^−3^	4.2 × 10^−3^	4.2 × 10^−3^	4.0 × 10^−3^
Spinal cord	NTCP (%)	3.0 × 10^−5^	2.5 × 10^−5^	1.8 × 10^−5^	5.3 × 10^−6^	6.5 × 10^−6^	9.4 × 10^−6^	5.9 × 10^−6^

TCP, tumor control probability; NTCP, normal tissue control probability; PTV, planning target volume; 3D-CRT, three-dimensional conformal radiotherapy; IMRT, intensity modulated radiation therapy; VMAT, volumetric modulated arc therapy.

## Data Availability

The data presented in this study are available on request from the corresponding author.

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
