# Peer review of "Automatic Radiobiological Comparison of Radiation Therapy Plans: An Application to Gastric Cancer"

_cancers, 2022, doi:10.3390/cancers14246098_

Round 1
Reviewer 1 Report
The manuscript has significant content and provides a new tool to calculate the TCP and NTCP values useful for the radiobiological evaluation. The methods and results are clearly presented and conclusions are well supported.
Author Response
Response : No comments requiring any change in the manuscript were made by the Reviewer 1.
Reviewer 2 Report
This linear and well-constructed article aims to radiobiologically compare three treatment plans performed on two phantoms simulating adult male and female patients with gastric cancer, using a new dedicated software.
The construction of the tool and its validation are well explained, the reporting that is produced is clear and usable.
The results shown are comprehensive.
Discussion and conclusions are pertinent.
The tables and figures accompanying the work are clear with good captions and correctly inserted in the text.
The bibliography is rich, relevant and up-to-date.
The work could be enriched with an additional note about the possibilities of a routine use of the instrument and about the way the authors imagine its future diffusion to other centers interested in its use.
Overall the article is well structured, the topic is clearly illustrated: therefore it can be published.
Author Response
RESPONSE TO REVIEWER 2 COMMENTS
Point 1: The work could be enriched with an additional note about the possibilities of a routine use of the instrument and about the way the authors imagine its future diffusion to other centers interested in its use.
Response 1: We agree with the reviewer’s comment that these additions may enrich the manuscript. The following changes were made.
We replaced the first sentence of the second paragraph of the Discussion of the previously submitted manuscript “The current study proposed a new software tool enabling the direct, quick and automatic calculation of TCP and NTCP values from treatment plans.” with “This study introduced a new software tool for calculating the TCP and NTCP from radiation therapy plans.” The new text appears in the first sentence of the second paragraph of the Discussion section of the revised manuscript.
We also added the following text at the end of the second paragraph of the Discussion section of the revised manuscript: “The newly developed software tool may be routinely used as a clinical aid in the radiobiological comparison of any treatment plans. The user intervention in this process is minimal. The tool may directly give quick and automatic calculations of TCP and NTCP values. Further research is needed to investigate the application of the software tool to centers equipped with treatment planning systems different from that used in this work.”